# Molecular and Cellular Mechanism of Action of Chrysotile Asbestos in MRC5 Cell Line

**DOI:** 10.3390/jpm13111599

**Published:** 2023-11-12

**Authors:** Assiya Kussainova, Olga Bulgakova, Akmaral Aripova, Milana Ibragimova, Alessandra Pulliero, Dinara Begimbetova, Rakhmetkazhi Bersimbaev, Alberto Izzotti

**Affiliations:** 1Department of Health Sciences, University of Genova, Via Pastore 1, 16132 Genoa, Italy; assya.kussainova@gmail.com (A.K.); alessandra.pulliero@unige.it (A.P.); 2Department of General Biology and Genomics, Institute of Cell Biology and Biotechnology, L.N. Gumilyov Eurasian National University, Astana 010008, Kazakhstan; ya.summer13@yandex.kz (O.B.); aripova001@gmail.com (A.A.); milanaibragimova2602@yandex.ru (M.I.); ribers@mail.ru (R.B.); 3National Laboratory Astana, Nazarbayev University, Astana 010000, Kazakhstan; dinara.begimbetova@nu.edu.kz; 4Department of Experimental Medicine, University of Genoa, 16132 Genoa, Italy; 5IRCCS Ospedale Policlinico San Martino, 16132 Genoa, Italy

**Keywords:** chrysotile asbestos, reactive oxygen species, cf mtDNA, microRNA, lung cancer

## Abstract

Asbestos is a known carcinogen; however, the influence of chrysotile asbestos on the development of tumor-related diseases remains a subject of intense debate within the scientific community. To analyze the effect of asbestos, we conducted a study using the MRC5 cell line. We were able to demonstrate that chrysotile asbestos stimulated the production of reactive oxygen species (ROS), leading to cell death and DNA damage in the MRC5 cell line, using various techniques such as ROS measurement, comet assay, MTT assay, and qPCR. In addition, we found that chrysotile asbestos treatment significantly increased extracellular mitochondrial DNA levels in the culture medium and induced significant changes in the expression profile of several miRNAs, which was the first of its kind. Thus, our research highlights the importance of studying the effects of chrysotile asbestos on human health and reveals multiple adverse effects of chrysotile asbestos.

## 1. Introduction

Despite the positive trend of a decrease in the number of new cases of lung cancer in the later stages and an increase in the life expectancy of cancer patients, the total number of new cases of lung cancer in the early stages remains very high [1]. According to the Kazakh Research Institute of Oncology and Radiology, lung cancer in the Republic of Kazakhstan ranks second in the structure of cancer diseases in general and accounts for 11.1% of all cases in the republic [2]. According to The Surveillance, Epidemiology, and End Results (SEER), in 2023 the global incidence of lung cancer will be 12.2% of all new cancer cases [3].

Lung cancer is known to be a multifactorial disease based not only on genetic aspects but also on environmental factors. The International Agency for Research on Cancer (IARC) has identified tobacco smoke, radon, and asbestos as major carcinogenic risk factors for lung cancer [4]. Prolonged exposure to these carcinogens often leads to the development of precancerous diseases. For example, prolonged exposure to asbestos dust leads to asbestosis [5].

Asbestosis is a chronic lung disease in which lung tissue is gradually replaced by fibrous tissue as a result of exposure to asbestos fibers. Asbestosis is a form of pulmonary fibrosis and belongs to the class of interstitial lung diseases [6].

Asbestos is a fibrous mineral belonging to the silicate class. Asbestos fibers, depending on their shape, are presented in two types: serpentine and amphibole. Serpentine fibers include chrysotile. Peridot is considered a less toxic form of asbestos fiber due to its sinuous shape. When inhaled, such fibers preferentially settle in the upper respiratory tract, where the mucociliary system of the respiratory tract is more pronounced and peridot fibers are easily excreted from the body. In turn, amphibole asbestos fibers (crocidolite, amosite, tremolite, and anthophyllite) are more toxic; they have a straight shape and penetrate the epithelial tissue of the bronchi like needles [7]. All types of asbestos fibers are recognized by the IARC as Group 1 carcinogens.

Exposure to asbestos occurs in three ways. First, asbestos dust from mining operations provides direct exposure. Second, workers in the construction and automotive industries are typically exposed indirectly. Third, there is exposure to the environment: asbestos fibers can contaminate the environment, for example, in building materials, and ultimately expose the general public.

Due to its structure, asbestos has high fire resistance and electrical and thermal insulation; this material has been widely used in various industrial spheres. Currently, there are few industries that do not use some of asbestos’s valuable properties. However, by 2019, asbestos was banned in 66 countries and regions around the world [8]. The world asbestos production in 2020 was more than 1.2 million tons. The main producing countries are Russia, Kazakhstan, China, and Brazil [9]. In terms of chrysotile asbestos deposits, Kazakhstan ranks fourth in the world. The only producer of chrysotile fiber in Kazakhstan is Kostanay Minerals JSC. During its 55 years of operation, the company has produced more than 17 million tons of chrysotile, which is exported to 20 countries [10].

Fibers less than 3 μm in diameter (chrysotile 0.026 μm) have the highest probability of entering the alveolar space and being cleared by surfactant proteins or macrophages [11]. Asbestos fibers that are too long penetrate into the cells, where they interfere with phagocytosis and trigger the primary production of reactive oxygen species (ROS). ROS production is primarily mediated by the reduction of Fe^3+^ iron ions on the surface of asbestos fibers [12]. Active cellular inflammation, in which macrophages initiate a cascade of biochemical reactions, results in secondary ROS production, a cascade of biochemical reactions, culminating in inflammation (Figure 1). Gualtieri outlines the diverse cellular impacts of asbestos exposure (including chrysotile), which include the formation of reactive oxygen species (ROS), the release of growth factors (TGF-β), the activation of p53, the activation of the Nalp3 inflammasome, the release of cytokines (TNF-α), the activation of transcription factors (AP-1, NF-κB), and the production of receptor tyrosine kinases (RTK) [13]. One consequence of oxidative stress is mtDNA release [14], which initiates a series of inflammatory responses leading to the development of specific diseases such as lung cancer [15].

Long asbestos fibers penetrate the macrophage, leading to oxidative stress. Macrophages are not able to phagocytose such fibers, and asbestos penetrates into other cells. Exposure to asbestos fibers on epithelial cells leads to the development of lung cancer, on fibroblasts to fibrosis, and on mesothelial cells to mesothelioma [15].

Asbestos is known not only to damage cellular structures by inducing oxidative cell stress but also to induce a wide range of changes at molecular and epigenetic levels (microRNA profiling or methylation) [16].

MicroRNAs (miRNAs) are short non-coding RNAs that regulate gene expression. Numerous studies have shown that the profile of microRNA changes due to various environmental hazards, including asbestos [17]. By regulating the expression of various genes, microRNAs are able to control many cellular processes that can lead to malignant cell transformation or, conversely, prevent it. The study of the microRNA profile is of practical importance because any changes in their expression may be associated with the development of fibrosis and lung cancer later. For example, the expression levels of miR-197-3p in the serum of workers who were exposed to asbestos were lower compared with controls. miR-197-3p may be a potential biomarker for mesothelioma development, along with chest X-ray, computed tomography, and spirometry [18]. Accordingly, it is interesting to study the microRNA pool as biological markers for early asbestos-related diseases.

## 2. Materials and Methods

### 2.1. Cultivation of Cell Culture 

MRC5 cells were cultured in Dulbecco’s Modified Eagle Medium (DMEM) (Capricorn Scientific GmbH, Ebsdorfergrun, Germany) with high glucose content, L-glutamine, 100-unit penicillin-streptomycin (Capricorn Scientific GmbH, Ebsdorfergrun, Germany), and 10% Fetal Bovine Serum (FBS), collected in South America (Capricorn Scientific GmbH, Ebsdorfergrun, Germany) in a humidified incubator at 37 °C with 5% CO_2_.

The MRC5 cell line was obtained from Nazarbayev University.

### 2.2. Treatment of MRC5 Cells with Asbestos

Chrysotile asbestos was produced at the asbestos mining and processing plant Kostanay Minerals JSC (Zhitikara, Kazakhstan). For all experiments, chrysotile asbestos was washed and resuspended in phosphate buffer and autoclaved. To obtain a homogeneous suspension, asbestos was crushed using a Bioruptor^®^ Plus sonication device at a purity of 20 kHz for 30 s for three cycles. To assess the effects of asbestos on MRC5, we added microdoses of asbestos at 1, 2.5, 5, 10, and 50 µg/cm^2^ of surface area 48 h after trypsinization.

### 2.3. ROS Measurement

The cells were seeded into a 96-well plate at 5 × 10^3^ cells/well. The fluorescent dye CM-H2DCFDA (#C6827, Thermo Fisher Scientific, Waltham, MA, USA) was used to detect the AFC levels in cells. CM-H2DCFDA was dissolved in 34.6 µL dimethyl sulfoxide (DMSO). The cellular suspension was re-absorbed with a 1:9 ratio CM-H2DCFDA solution and incubated at 37 °C for 30 min. After incubation, the AFC level was measured using the BioTek Cytation 5 Cell Imaging Multimode Reader (Agilent Technologies, Santa Clara, CA, USA) multipurpose microplate reader for excitation at 494 nm and emission at 522 nm after 1, 3, 6, 12, and 24 h of exposure to chrysotile asbestos. Hydrogen peroxide was used as a positive control (at a final concentration of 0.5 mM).

### 2.4. Comet Assay

MRC5 cells were seeded into 6-well plates at a rate of 5 × 10^6^ cells/well. The cells were incubated with chrysotile asbestos for 24 h. A comet assay was carried out according to the protocol described in the article by Vandghanooni [19] with some changes. SYBR Green was used to color the comets.

The alkaline comet assay was a sensitive test for the detection of single- and double-strand DNA breaks in individual cells. As the frequency of DNA breaks increased, so did the comet tail, which consisted of the fraction of genomic DNA that had been exposed to ROS. The degree of DNA damage (single- and double-strand breaks) was determined as the percentage of DNA fluorescence in the tail (tail DNA%), which was calculated using the “Tri Tek Comet Score version 1.5” software.

### 2.5. Determination of Cell Viability

The cell sensitivity to asbestos was assessed using the MTT assay. Cells were seeded into a 96-well plate at a rate of 5 × 10^3^ cells/well and incubated for 24 h at 37 °C. Then asbestos was added to the wells in various concentrations and incubated for another 12, 24, and 48 h at 37 °C. The cell viability was measured using the MTT Cell Viability Assay Kit (#30006, Biotium, Inc., Fremont, CA, USA) according to the manufacturer’s protocol. The absorbance signal was measured using a spectrophotometer BioTek Cytation 5 Cell Imaging Multimode ReaderCytation (Agilent Technologies, Santa Clara, CA, USA) at a wavelength of 570 nm. As a positive control, the cells were treated with DMSO.

### 2.6. Isolation of cf mtDNA

After culturing cells with and without asbestos, the medium was collected after 4, 24, and 48 h. To isolate total cell-free circulating DNA, a commercial PROBA-NK reagent kit (#D07-2, DNA-Technology, Moscow, Russia) was used in accordance with the manufacturer’s protocol.

### 2.7. cf mtDNA Isolation from Medium

Mitochondrial 16S RNA with a fragment length of 230 nucleotides was chosen for amplification, which makes it possible to determine the number of copies of mtDNA released from cells as a result of cell death [20].

Specific primers were designed for the 230 bp fragment as follows: forward primer, 5′-CAGCCGCTATTAAAGGTTCG-3′; reverse, 5′-GGGCTCTGCCATCTTAACAA-3′.

For the preparation of standards, PCR of some samples was performed. The reaction mixture included PCR Master Mix (2X) (#K0171, Thermo Fisher Scientific, Waltham, MA, USA), 20 pmol of forward/reverse primer, and 100 ng of DNA sample. The program was set to 95 °C for 10 min and 40 cycles (95 °C for 15 s, 60 °C for 1 min, and 72 °C for 1 min 15 s). For detection of a 230 bp fragment, electrophoresis was performed in 1.5% agarose gel using a DNA marker (GeneRuler DNA Ladder Mix, Thermo Fisher Scientific, Waltham, MA, USA). The QIAquick PCR Purification Kit (#28104, Qiagen, Hilden, Germany) was used to purify the PCR product according to the manufacturer’s protocol. The DNA concentration was determined using Quant-iT™ PicoGreen™ dsDNA Assay Kits (#P11496, Thermo Fisher Scientific, Waltham, MA, USA) according to the manufacturer’s protocol.

The copy number was determined by using real-time PCR. The composition of the reaction mixture for real-time PCR was as follows: Maxima SYBR Green/ROX qPCR Master Mix (2X) (#K0222, Thermo Fisher Scientific, Waltham, MA, USA), 20 pmol of forward/reverse primer, and 100 ng of DNA sample.

The PCR protocol included the following steps: heating at 90 °C for 10 min, then 40 cycles with denaturation at 95 °C for 15 s, and termination at 60 °C for 1 min. All experiments were performed using the QuantStudio™ 3 Real-Time PCR System in duplicate.

The calculation of the cf mtDNA copy number was performed based on a standard curve created according to the protocol “Creating Standard Curves with Genomic DNA or Plasmid DNA Templates for Use in Quantitative PCR” from Thermo Fisher Scientific. This method enables determining the exact number of copies of mitochondrial DNA in samples and provides more reliable research results.

### 2.8. microRNA Isolation from Cells

The cells were cultured in a 12-well plate at a rate of 5 × 10^5^ cells per well with the addition of asbestos. Cells were removed after 4, 24, and 48 h using Nunc™ Cell Scrapers (Thermo Fisher Scientific, Waltham, MA, USA). The microRNAs were isolated using the commercial miRNeasy Tissue/Cells Advanced Micro Kit (#217684, Qiagen, Hilden, Germany) according to the manufacturer’s protocol. The quantity and purity of the extracted RNA were assessed using a NanoDrop™ 2000 fiber optic spectrophotometer (#ND-2000, ThermoFisher Scientific, Inc., Waltham, MA, USA) in accordance with the manufacturer’s protocols; samples with a value higher than 1.85 at a coefficient of absorption 260/280 were identified.

### 2.9. microRNA Analysis by qPCR

cDNA was obtained from isolated miRNA samples using the commercial miRCURY LNA RT Kit (#339340, Qiagen, Hilden, Germany) according to the manufacturer’s recommendations. Real-time PCR was then performed on a QuantStudio™ 3 Real-Time PCR System in duplicate. Each reaction included a reaction mixture from miRCURY LNA SYBR^®^ Green PCR Kits (#339347, Qiagen, Hilden, Germany), specific primers, and cDNA samples. The PCR program was used: 95 °C for 10 min and 40 cycles (95 °C for 15 s and 56 °C for 1 min). All samples were amplified in triplicate, and the relative quantification of the expression level of each gene was calculated. U6 was used as the endogenous reference gene. To determine the level of microRNA expression, the 2^−ΔΔCt^ method was used.

### 2.10. Statistical Data Analysis

The statistical data analysis was performed using GraphPad Prism 9 software (GraphPad Prism 9.5.1.733 for Windows, GraphPad Software, Boston, MA, USA). All data are presented as mean ± standard deviation. The differences between experimental groups relative to the control were assessed using one-way ANOVA (for a normal sample) or the Kruskal–Wallis H test (for a non-normal sample). Values of *p* ≤ 0.05 were considered statistically significant (* *p* ≤ 0.05; ** *p* ≤ 0.01; *** *p* ≤ 0.001; **** *p* ≤ 0.0001; ns *p* > 0.05).

## 3. Results

### 3.1. Asbestos Increases ROS Production in MRC5

One of the main damaging mechanisms of asbestos is the induction of oxidative cellular stress. As a result, damage to the cellular components and DNA occurs. To determine whether chrysotile asbestos stimulates oxidative cellular stress, we examined the ROS levels based on the fluorescent signal of CM-H2DCFDA, as shown in Figure 1A.

Measurements of the cellular ROS levels 1 and 3 h after the start of the asbestos treatment for MRC5 showed no changes. Active production of ROS was observed after 6 h of exposure to asbestos at doses of 5 µg/cm^2^ and 10 µg/cm^2^. The level of ROS increased by 1.6 (*p* = 0.0034) and 2.1 (*p* = < 0.0001) times compared with the control. A similar effect was observed when the exposure time to asbestos was increased to 12 h. Doses of 5 µg/cm^2^ and 10 µg/cm^2^ stimulated the production of ROS. The level of ROS in the cells increased by 2.68 (*p* = 0.0019) and 3.70 (*p* = < 0.0001) times. The ROS levels were not affected by exposure to a dose of 2.5 µg/cm^2^ for 6 and 12 h (*p* < 0.4973, *p* < 0.2638). The maximum concentration of ROS was observed after 24 h of exposure to chrysotile. We found a twofold increase in ROS levels after exposure to 2.5 µg/cm^2^ chrysotile asbestos (*p* < 0.0001). Increasing the exposure dose to 5 µg/cm^2^ and 10 µg/cm^2^ increased the ROS production by 3.63 and 5.48 times, respectively (*p* < 0.0001) (Figure 1B).

Our results showed that oxidative cellular stress caused by the action of different doses of chrysotile asbestos had a dose-dependent effect. The higher the concentration of asbestos and the longer the exposure, the higher the level of ROS in the cells (Figure 2). A dynamic increase in the cellular ROS was observed at specified intervals. The maximum concentration of the cellular ROS was observed after 24 h of exposure to different doses of asbestos.

### 3.2. Genotoxic Effects of Chrysotile Asbestos

In most cases, single- and double-strand DNA breaks are caused by ROS. As previously shown, chrysotile asbestos can cause oxidative cellular stress. Already after 24 h of exposure, active ROS production was observed. As expected, after treating the cells with different doses of chrysotile asbestos for 24 h, nuclei with comet tails were detected (Figure 3A).

In MRC5 cells, the percentage of tail DNA increased as a function of increasing chrysotile asbestos dose, from 9.87% at a dose of 2.5 µg/cm^2^, 27% at a dose of 5 µg/cm^2^, and up to 38% at a dose of 10 µg/cm^2^, while the average level of the tail DNA without the addition of asbestos was less than 1% (Figure 3B). The tail DNA% levels were higher when the cells were treated with chrysotile asbestos, suggesting that asbestos may cause oxidative DNA damage.

### 3.3. Cytotoxic Effect of Chrysotile Asbestos on Healthy Lung Fibroblasts

The cytotoxicity potential of chrysotile asbestos was assessed using the MTT assay. The percentage of cell survival was assessed by adding various doses of asbestos fibers with exposures of 6, 12, 24, and 48 h. According to the obtained results, it can be assumed that the cytotoxicity of asbestos increased with increasing concentration and exposure time. The treatment of cells with asbestos for 6 and 12 h did not show any obvious cytotoxic effect. Asbestos showed minimal cytotoxicity potential at 5 µg/cm^2^ after 24 h of exposure (*p* = 0.0039). And as can be seen in the graphs (Figure 4), the higher the dose of asbestos and the longer the period of exposure, the lower the percentage of viable cells. Increasing the exposure dose to 10 µg/cm^2^ led to a decrease in cell survival to 75.8% (*p* < 0.0001). A significant dose-dependent effect of asbestos was observed after 48 hours, when the percentage of viable cells was 83%, 74 and 65.9% when exposed to corresponding doses of asbestos fibers. The cytotoxicity of chrysotile asbestos increased with increasing dose and exposure time, which proved that asbestos had a dose-dependent cytotoxic effect (Figure 4).

### 3.4. The Copy Number of cf mtDNA in Media

Oxidative stress leads to damage to cellular structures and can be a significant cause of cell death. cf mtDNA can be not only secreted by cells but also released into the extracellular space as a result of cell death. It is assumed that this fragment is 230 bp is a product of apoptotic cell death [21]. And as a consequence, fragments of this length should be found in the extracellular space, i.e., the nutrient medium in the case of cell cultures. Extracellular free-circulating mtDNA was determined in a nutrient medium after culturing cells for 4, 24, and 48 h with the addition of various doses of chrysotile asbestos (2.5, 5, and 10 µg/cm^2^).

We found an increase in the amount of extracellular mtDNA when cells were treated with 10 µg/cm^2^ after 4 h of exposure to chrysotile asbestos; the median of cf mtDNA was 3.54 × 10^12^ copies/mL (*p* = 0.0096). The treatment of cells with asbestos doses of 2.5 and 5 µg/cm^2^ did not show any significant differences, with *p* = 0.9220 and *p* = 0.2666, respectively (Figure 5). A significant increase in the amount of cf mtDNA after 24 h of exposure to asbestos dust, a median of cf mtDNA of 9.72 × 10^14^ copies/mL (*p* = 0.0067), was shown at an exposure dose of 10 µg/cm^2^. An increase in the cf mtDNA copy number was observed at doses of 2.5 and 5 µg/cm^2^, but this was not statistically significant (*p* = 0.9245 and *p* = 0.1246) (Figure 5). An increase in mtDNA copies was found after 48 h of cell treatment with various doses of asbestos; the median of cf mtDNA at a dose of 2.5 µg/cm^2^ was 1.47 × 10^15^ copies/mL (*p* = 0.0393), at a dose of 5 µg/cm^2^ the median of cf mtDNA was 4.49 × 10^15^ copies/mL (*p* = 0.0001), and at a dose of 10 µg/cm^2^ the median of cf mtDNA was 1.37 × 10^16^ copies/mL (*p* ≤ 0.0001) (Figure 5).

Our results clearly showed that the longer the exposure time to chrysotile, the higher the number of extracellular mtDNA copies. Moreover, a sharp increase in cf mtDNA was observed after 48 h of incubation (Figure 6).

### 3.5. MicroRNA Expression Profile Analysis between Asbestos-Treated Cells and Controls

We analyzed the literature data, after which we selected several microRNAs whose profiles could be changed in response to the action of chrysotile asbestos. The exposure times to asbestos dust were selected as 4, 24, and 48 h. There is evidence that the microRNA profile can change after just 2 h of exposure [22]. We analyzed the expression levels of six microRNAs (hsa-miR-19b-3p, hsa-miR-125b-5p, hsa-miR-181b-5p, hsa-miR-376b-3p, hsa-miR-1202, and hsa-miR-1228) at different intervals when exposed to different doses of chrysotile asbestos.

We found that hsa-miR-181b-5p expression was suppressed after 4 h at a chrysotile asbestos exposure dose of 2.5 µg/cm^2^. A change in the expression levels of two microRNAs, namely, hsa-miR-181b-5p and hsa-miR-1202, was found after 24 h of exposure to chrysotile at doses of 2.5 µg/cm^2^ and 10 µg/cm^2^, respectively. The expression profile of almost all selected miRNAs was altered after 48 h of exposure to different doses of chrysotile asbestos. An increase in the expression of hsa-miR-19b-3p, hsa-miR-376b-3p, and hsa-miR-1202 was observed at a dose of 5 µg/cm^2^. The level of hsa-miR-181b-5p was reduced at a dose of 2.5 µg/cm^2^, but the level of hsa-miR-125b-5p, on the contrary, increased. Moreover, hsa-miR-125b-5p showed a change in the expression profile not only at 2.5 µg/cm^2^ but also at 10 µg/cm^2^. We did not find any significant differences in the expression of hsa-miR-1228 after treating cells with different doses of chrysotile asbestos at specified time intervals (Figure 7).

By observing how the relative expression of these microRNAs changes, some conclusions can be drawn. The change in the expression of hsa-miR-181b-5p after 4 h of exposure to asbestos suggested that this microRNA belonged to the molecules of the emergency cellular response. The expression profile of other microRNAs, except hsa-miR-181b-5p and hsa-miR-1228, was changed only after 48 h of exposure. In general, it was difficult to trace the pattern of changes in microRNA expression depending on the dose and time of exposure to chrysotile asbestos. The microRNA expression changed at different times when exposed to different doses of asbestos. This may have been due to processes regulated by microRNAs. In any case, this requires further study.

## 4. Discussion

Lung diseases associated with asbestos exposure remain an important global health problem. Italy, Canada, and Russia were the largest miners of asbestos between 1866 and 1890. Because asbestos is a versatile material, it has been actively used since the 1950s in the production of a wide range of products used in everyday life. The asbestos industry peaked in the 1970s, after which the rate of mass use of asbestos began to decline [23]. If we take into account that the incubation period of asbestos fibers reaches up to 40 years, it turns out that only in the last 20 years have we fully experienced the effects of the asbestos industry on public health [24].

Today, 52 countries around the world have banned the use of asbestos [25]. But despite all the known information about the toxicity and danger of asbestos fibers, asbestos is mined, produced, and exported in many countries.

Different forms of asbestos have varying degrees of toxicity to the body: crocidolite is 500 times more toxic than chrysotile. Chrysotile asbestos, due to its structure, is considered the safest form of asbestos fiber. Currently, 95% of asbestos used and produced is chrysotile [26].

All forms of asbestos fibers (amphibole and serpentine) are known to be associated with a high risk of developing lung cancer and mesothelioma. Accordingly, the absence of a ban on the use of asbestos will lead to an increase in the incidence of these diseases. The cumulative effect of asbestos and smoking increases the risks of not only morbidity but also mortality from these forms of cancer [27].

Chrysotile asbestos is readily available and widely used in the construction industry because it is considered the least harmful form of asbestos fiber. It is available not only on an industrial scale but also in mass markets. In countries where asbestos is not banned, chrysotile is a preferred material due to its affordability.

Research suggests that this type of asbestos fiber can have harmful effects on cellular functions, contrary to the belief that chrysotile asbestos is safe for human health. Several mechanisms have been identified for the cellular response to asbestos. The infiltration of asbestos fibers into cells leads to increased levels of reactive oxygen species (ROS). This causes oxidative stress, which in turn damages cellular components and DNA. As a result, a large number of signaling systems may be activated, leading to cell death or fibrosis and, subsequently, to carcinogenesis [28].

The analysis of the literature data showed that different cell cultures exhibited different sensitivity to the effects of asbestos (Table 1). The range of chrysotile concentrations within which molecular and cellular effects were observed ranged from 1 to 150 µg/cm^2^.

For our study, we decided to use minimal doses of chrysotile, which could induce a certain cellular response after the treatment of human lung fibroblasts (MRC5). Preliminary experiments showed that at a dose of 1 µg/cm^2^, no cellular response to the action of chrysotile was observed. A dose of 50 µg/cm^2^ was too cytotoxic. Therefore, all experiments were carried out using chrysotile in the following concentrations: 2.5, 5, and 10 µg/cm^2^.

We showed a clear dose-dependent effect of chrysotile; the higher the dose and the longer the exposure time to asbestos fibers, the more pronounced was the observed cellular response. The minimum dose at which cellular changes were observed was 2.5 µg/cm^2^.

The intercalation of asbestos fibers into MRC5 induced the production of cellular ROS. Crocidolite induced an increase in ROS within 30 min of cell treatment and a further increase in ROS within 12 h [28]. A change in ROS levels after the treatment of cells with chrysotile occurred only after 6 h of incubation. An increase in cellular ROS levels was observed within 24 h. Considering that single- and double-stranded DNA breaks were detected as a result of chrysotile exposure, it could be concluded that chrysotile induced cellular oxidative stress resulting in DNA oxidation. However, no correlation was found between cellular ROS levels and DNA damage.

The repair of single- and double-strand DNA breaks is a complex process; cells strive to survive and activate the work of repair systems. However, translocations, duplications, and deletions of genes that occur during repair can contribute to malignant cell transformation and tumor development [36]. To prevent a cell from degenerating into a cancerous one, signaling pathways are activated that seek to remove the damaged cell from the pool of healthy ones, triggering programmed cell death. Signaling triggered by DNA damage is specific to different cell types and also depends on several intracellular factors, for example, the type of damaging agent, the p53 protein status, the activation of death receptors, signaling pathways, etc. [37]. It is known that the effect of chrysotile asbestos initiates cell apoptosis, initiating the release of cytochrome c [33], and increases the expression of effector kinases [38] and pro-inflammatory cytokines [39]. Our data showed a decrease in cell viability after only 24 h of incubation with chrysotile, and a minimal exposure dose of 2.5 µg/cm^2^ caused cell death after 48 h. No correlation was found between the increase in cellular ROS and the viability of the cells. However, cell death due to oxidative stress was suggested by the observation of free radical formation leading to genomic DNA breaks.

In addition to assessing cell viability, we examined the number of copies of extracellular mtDNA in a nutrient medium where cells were cultured with the addition of chrysotile. The accumulation of damage in mitochondria that was caused by oxidative stress leads to impaired respiratory function, which ultimately leads to oxidation of mtDNA [40]. As a result of mitochondrial degradation, mtDNA can be released into the cytosol and activates a wide range of cellular responses [41]. Oxidative stress has been shown to release mtDNA fragments smaller than 700 bp [42]. As our data showed, the amount of extracellular mtDNA increased with the duration of exposure to chrysotile. Maximum copy numbers were detected after 48 h of exposure to all selected concentrations of asbestos fibers. In general, there was a relationship between the level of cell viability and the number of copies of extracellular mtDNA, but no correlation was found between these parameters.

The precise processes underlying the release of mtDNA are not yet fully understood. However, there is evidence to suggest that following mitophagy, the cell releases the mtDNA that has accumulated in the cytosol. Apoptosis or necrosis that damages the cell membrane is also likely to result in the release of mtDNA [43]. The released ccf-mtDNA enters other cells via endocytosis and induces inflammation through the activation of pro-inflammatory signaling pathways [44].

Asbestos concentrations of 2.5, 5, and 10 µg/cm^2^ may appear to be low exposure doses, and it may come as a surprise that they are capable of causing significant effects at the cellular and molecular levels. However, when these concentrations are calculated on the basis of the internal lung surface area (ISA) of an adult, the concentrations chosen are actually quite high. Considering an average ISA of 45 m^2^ [45], a dose of 2.5 µg/cm^2^ is equivalent to 1.125 g of asbestos inhaled into the lungs. Therefore, even seemingly small concentrations of asbestos can have a considerable impact on respiratory health. Therefore, doses of 5 and 10 µg/cm^2^ equate to 2.25 g and 4 g, respectively. It has been found that workers in the asbestos mining industry inhale lesser amounts of asbestos throughout the year [46]. Overall, the acute form of exposure to chrysotile was modeled and investigated in our study.

Not only mining industry employees are exposed to asbestos but also manufacturing workers and the general population. Many studies have confirmed the connection between exposure to chrysotile asbestos, asbestosis, and lung cancer [47,48]. Screening the population for lung cancer is particularly difficult because there is currently no clear biological marker. As a result, the diagnosis of lung cancer is often only at a late stage, when the disease is already advanced. One promising marker for liquid biopsy is microRNAs. These molecules change their expression depending on many factors, including the action of external environmental factors [49]. microRNAs may well be used to monitor the state of the body under the influence of asbestos. After analyzing scientific publications, we selected six microRNAs that were associated with the development of fibrosis: hsa-miR-19b-3p, hsa-miR-125b-5p, hsa-miR-181b-5p, hsa-miR-376b-3p, hsa-miR-1202, and hsa-miR-1228.

The miR-19a-19b-20a sub-cluster suppresses TGF-β-induced fibroblast activation in vitro [50]. miR-125b-5p negatively regulates p53 protein expression in human neuroblastoma cells and lung fibroblasts [51]. The overexpression of miR-181b-5p inhibits apoptosis and promotes fibroblast proliferation by regulating the MEK/ERK/p21 signaling pathway [52]. But increasing the expression of miR-376b-3p, on the contrary, inhibits the proliferation of fibroblasts, causing their premature aging [53]. miR-1202 promotes fibrosis by inducing TGF-β1 [54]. According to our assumptions, chrysotile asbestos could cause changes in the expression of these microRNAs. Indeed, the profile of these microRNAs (with the exception of miR-1228) changed in response to the action of chrysotile fibers of different concentrations. Given the general trend in the alteration of microRNA profiles, it is not yet possible to draw any definite conclusions about a dose-dependent effect. At different exposure levels and after different periods of exposure to chrysotile, the expression of different microRNAs varies. The results provide further prospects for studying the profile of miRNA data at the population level.

## 5. Conclusions

In summary, the safety of asbestos mining and processing is a major concern for many countries, including Kazakhstan, which is a leading producer of asbestos. Chrysotile asbestos is generally considered to be a less dangerous form of asbestos and is not considered to be carcinogenic.

Our study in the MRC5 cell line showed that exposure to chrysotile asbestos leads to the production of free radicals, which induce oxidative cellular stress in a dose-dependent manner. This stress, as evidenced by the presence of single- and double-strand breaks in chrysotile-exposed cells, ultimately leads to DNA damage. In addition, apoptosis may be induced in the damaged cells. Our results indicate a decrease in cell viability after incubation with chrysotile asbestos for as little as 24 h. There is also a high concentration of extracellular mitochondrial DNA in the culture medium, indicating apoptotic cell death.

Notably, previous studies, including our own, have shown a strong correlation between levels of circulating mitochondrial DNA and the risk of developing cancer. In addition, we observed an altered expression profile of several microRNAs in surviving cells that regulate processes such as proliferation, apoptosis, and fibrosis. These results call into question the assertion that chrysotile asbestos is safe for human health as this type of fiber has the potential to induce adverse cellular effects at the molecular and epigenetic levels.

Further comprehensive population-based studies are needed to assess the impact of this type of asbestos fiber on human health. Based on our findings, there is an urgent need to re-evaluate the safety presumption for chrysotile asbestos and to emphasize ongoing research efforts to better understand its potential health hazards.

## Figures and Tables

**Figure 1 jpm-13-01599-f001:**
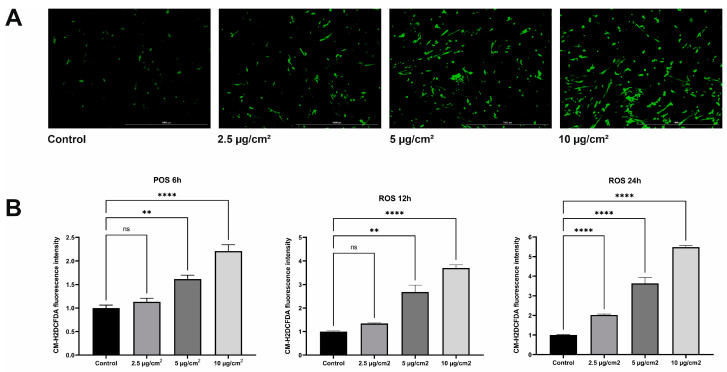
ROS detection by CM-H2DCFDA staining: (**A**) Fluorescence microscopic image of chrysotile asbestos-treated cells after 24 hours of exposure. The scale bar corresponds to 1000 µm; (**B**) Spectrophotometric measurement of fluorescence intensity of CM-H2DCFDA, which indicates an increase in ROS in cells treated with different doses of chrysotile asbestos. (** *p* ≤ 0.01; **** *p* ≤ 0.0001; ns *p* > 0.05).

**Figure 2 jpm-13-01599-f002:**
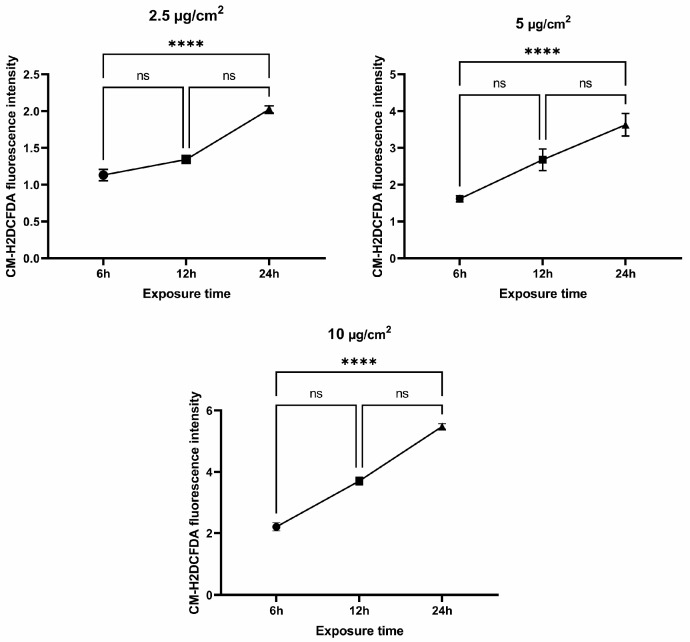
Dynamics of the growth of cellular ROS after 6, 12, and 24 h of exposure to chrysotile asbestos at doses of 2.5, 5, and 10 µg/cm^2^. (**** *p* ≤ 0.0001; ns *p* > 0.05).

**Figure 3 jpm-13-01599-f003:**
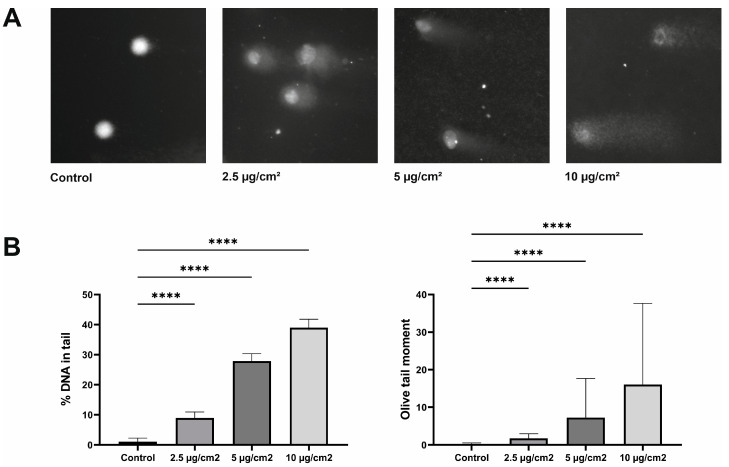
Evaluation of DNA damage by the comet assay: (**A**) MRC5 were treated with chrysotile asbestos for 24 h and used an alkaline comet assay to detect single- and double-strand DNA breaks (scale bar, 200 μm). Images were acquired using an Olympus fluorescence microscope (Shinjuku, Tokyo, Japan). (**B**) Graphic summary of comet assay results. Percentage of fragmented DNA in the tail and quantification of tail moment. (**** *p* ≤ 0.0001).

**Figure 4 jpm-13-01599-f004:**
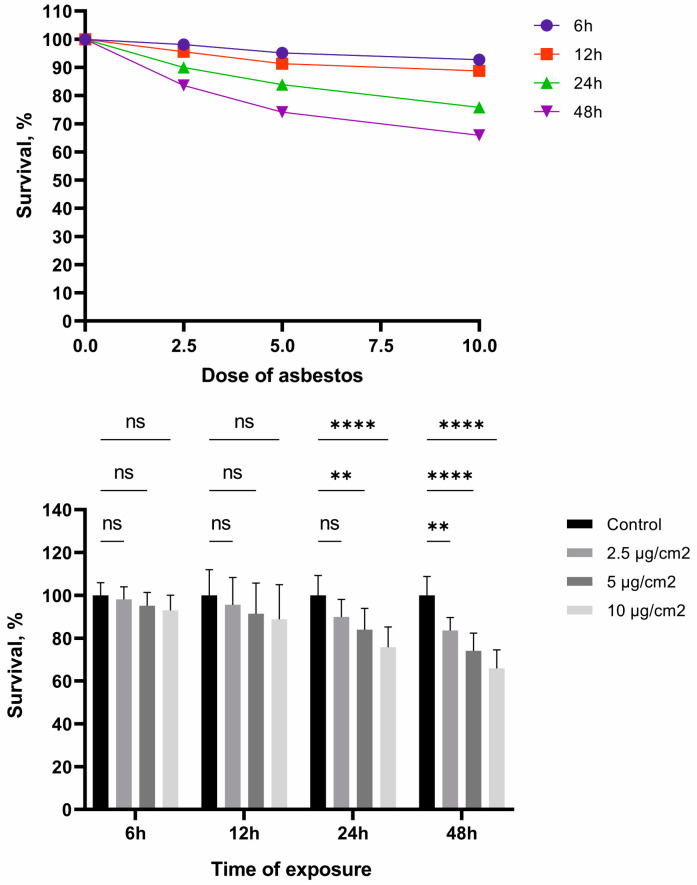
Cell viability assay results. A decrease in cell viability was shown after exposure to increasing concentrations of chrysotile asbestos for 6, 12, 24, and 48 h. The viability of MRC5 was significantly reduced after 24 h of exposure. (** *p* ≤ 0.01; **** *p* ≤ 0.0001; ns *p* > 0.05).

**Figure 5 jpm-13-01599-f005:**
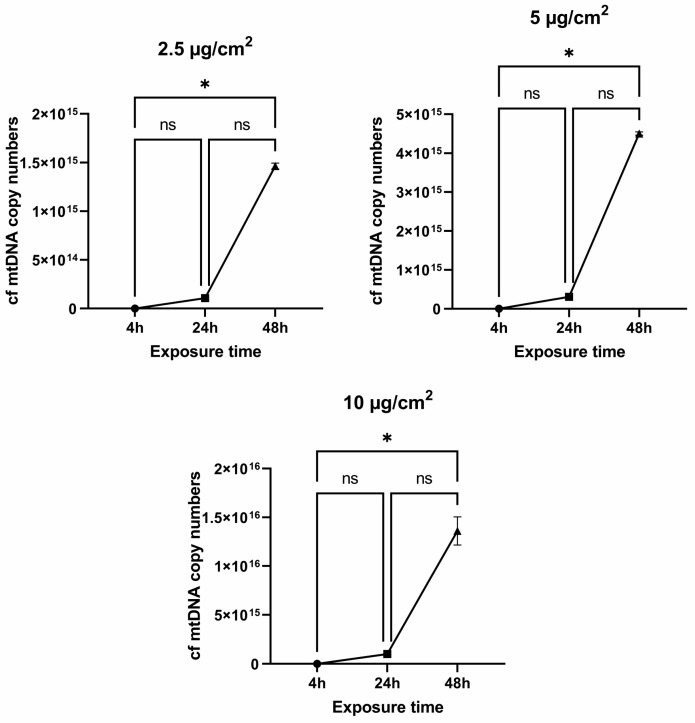
The cf mtDNA copy number in a medium where MRC5 were cultured with the addition of chrysotile asbestos at 2.5, 5, and 10 µg/cm^2^ for 4, 24, and 48 h. (* *p* ≤ 0.05; ns *p* > 0.05).

**Figure 6 jpm-13-01599-f006:**
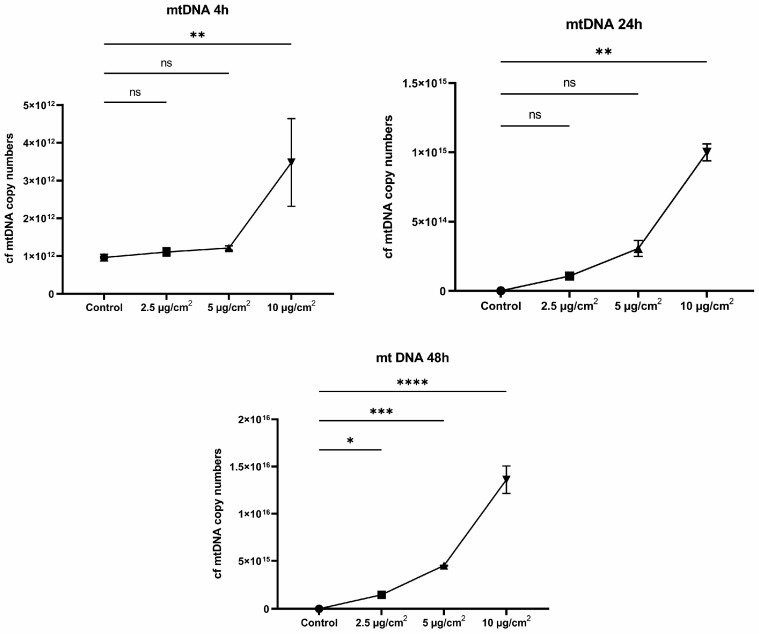
Dynamics of increase in the cf mtDNA copy number after 4, 24, and 48 h of exposure to chrysotile asbestos at doses of 2.5, 5, and 10 µg/cm^2^. (* *p* ≤ 0.05; ** *p* ≤ 0.01; *** *p* ≤ 0.001; **** *p* ≤ 0.0001; ns *p* > 0.05).

**Figure 7 jpm-13-01599-f007:**
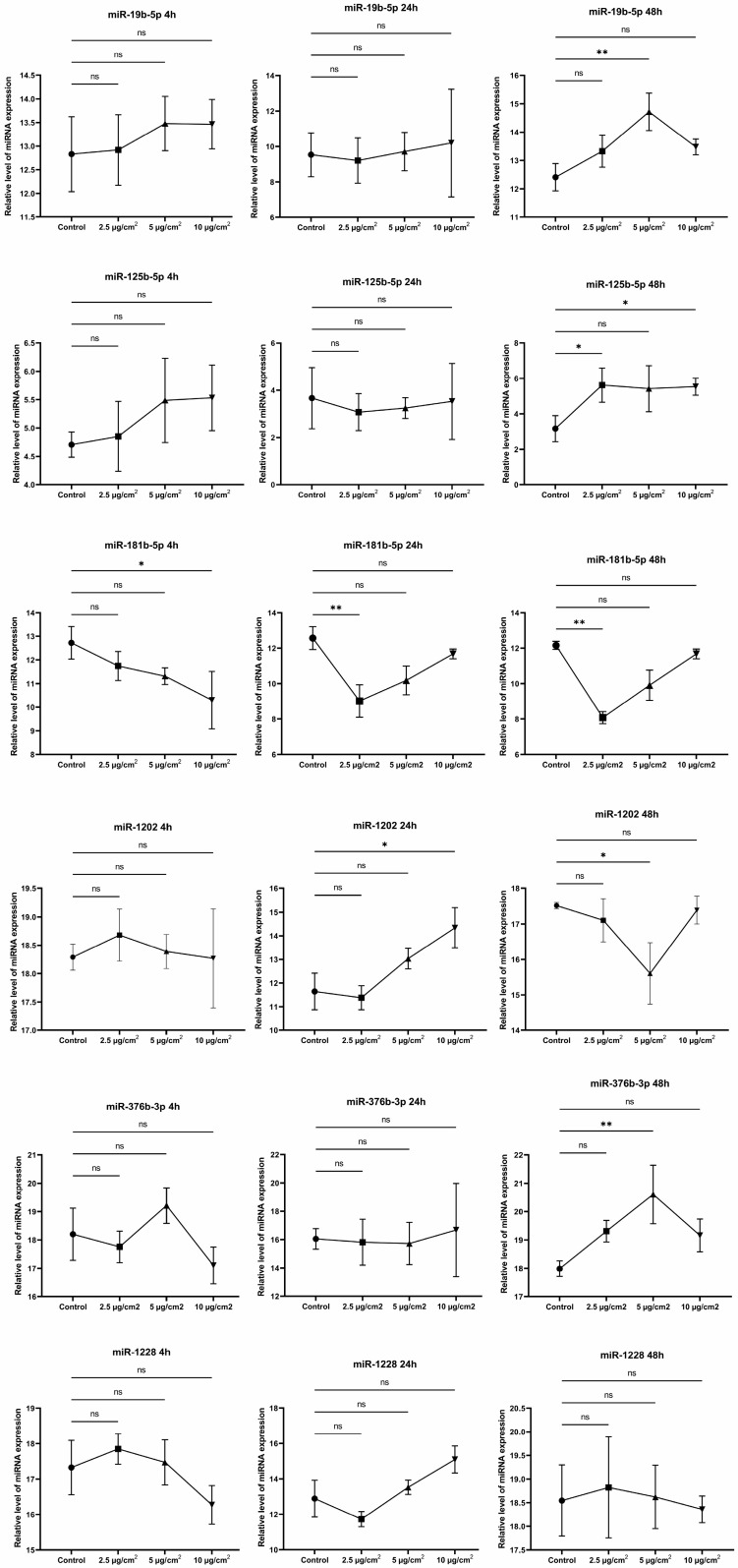
Graphical summary of the relative expression of hsa-miR-19b-3p, hsa-miR-125b-5p, hsa-miR-181b-5p, hsa-miR-376b-3p, hsa-miR-1202, and hsa-miR-1228 in MRC5 cells after 4, 24, and 48 h of exposure to chrysotile asbestos at doses of 2.5, 5, and 10 µg/cm^2^. (* *p* ≤ 0.05; ** *p* ≤ 0.01; ns *p* > 0.05).

**Table 1 jpm-13-01599-t001:** Comparative table of experimental asbestos exposure doses.

Dose of Exposure	Asbestos Fiber Type	Cell Culture	Exposure Time	Cellular Effect	Reference
1 µg/cm^2^	Chrysotile	BEAS-2B, NuLi-1, A549	30 min, 1, 3, 6, and 72 h	Significantly decreased levels of E-cadherin and β-catenin were noted, while TGF-β levels were increased. After 30 min, there was a decrease in cytoplasmic p-Smad2 and an increase in nuclear p-Smad2. TGF-β levels were increased after 1 h of asbestos exposure. The ratio of phosphorylated to nonphosphorylated Akt (p-Akt/Akt) was increased from 30 min to 3 h.	[29]
1 µg/cm^2^	Crocidolite	LP9 and HPM3	2 weeks	SNAI1 protein was shown to de-crease at 24 and 48 h in LP9 cells, and HPM3 cells showed a twofold increase after 1 week of asbestos exposure. TIMP1 protein was increased 36-fold in LP9 after just 24 h of exposure.	[30]
1, 5, 10, and 20 µg/cm^2^	Crocodilite	Murine peritoneal macrophages (MF)	24 h	An asbestos dose of 20 µg/cm^2^ caused a twofold increase in ROS, as well as an increase in nuclear Nrf2 levels, with the highest concentrations observed 2 and 12 h after exposure.	[28]
0.1–0.5 µg/cm^2^	Crocidolite	Cloned diploid hamster tracheal epithelial cells	3–24 h	Increases in [3H] thymidine incorporation and colony formation efficiency were observed.	[31]
5 µg/cm^2^	Chrysotile	MeT-5A cells	24 h	Decreased expression of microRNA-28 has been shown.	[32]
50, 100, 150, 200, and 300 µg/cm^2^	Chrysotile	A549		A dose of 150 µg/cm^2^ provoked the release of cytochrome c and an increase in the level of Bax/Bak and caspase-9 and, as a result, caused apoptosis after 48 h of exposure.	[33]
50 µg/cm^2^	Chrysotile	A549	48 h	There was an increase in the expression of cleaved caspase-3 and -9.	[34]
5 µg/cm^2^	Crocidolite and chrysotile	HM cells	24 h	Chrysotile has been observed to induce rapid cell death and increased release of HMGB1 and TNF-α.	[35]

## Data Availability

The datasets used and/or analyzed during the current study are available from the corresponding author on reasonable request.

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
