# Peer review of "Molecular and Cellular Mechanism of Action of Chrysotile Asbestos in MRC5 Cell Line"

_jpm, 2023, doi:10.3390/jpm13111599_

Round 1

Reviewer 1 Report

Comments and Suggestions for Authors

In this article titles “Mechanisms of toxic effects of chrysotile asbestos on normal human lung fibroblasts,” the authors study the potential adverse effects of chrysotile asbestos exposure using an in vitro culture model of normal human lung fibroblasts. The findings suggest that chrysotile asbestos increase ROS, DNA damage as well as cell death in the fibroblast cells. Several miRNAs were also modulated by the exposure to chrysotile asbestos. Overall, the study is well conceived, and the results support the conclusions. Some minor suggestions –

11)      The abstract could be better written.

22)      In lines 100-101, maybe the authors could summarize why the specific miRNAs were studied – are they known to contribute to lung carcinogenesis? Have they been shown to contribute to fibrosis in other asbestos models? Please explain in more detail why these specific miRNAs were studied.

33)      In future studies, they authors should consider quantifying cell viability using other methods (trypan blue/ fix,stain nuclei and count on cytation5) as well. Assays such as MTT and CCK8 measure the activity of certain enzymes in the cells which we assume to correlate to cell number – but some treatments themselves can induce the activity of these enzymes, therefore, additional validation could strengthen their data.

44)      Keeping with the viability assays, it would be good to look at additional longer time points as well – you might see a more profound effect on viability at later times.

Comments on the Quality of English Language

Overall language is good. But, certain statements, for eg; line 60-61 uses a double negative, which makes it hard to understand.

Author Response

Dear Reviewer,

Thank you for reviewing our manuscript and providing feedback on our work. We hope you find the revised version to be satisfactory and thank you again for your time and attention. We have taken your comments into consideration and made the necessary improvements to enhance the clarity and precision of our text. We would like to express our gratitude for your valuable input and highlight that we have made every effort to maintain the original meaning of the text.

11) The abstract could be better written.

We have completely rewritten the abstract, hopefully we have conveyed the meaning and idea of the article as accurately as possible.

22) In lines 100-101, maybe the authors could summarize why the specific miRNAs were studied – are they known to contribute to lung carcinogenesis? Have they been shown to contribute to fibrosis in other asbestos models? Please explain in more detail why these specific miRNAs were studied.

We report the association of these microRNAs with pulmonary changes in lines 474-480.

miR-19a-19b-20a sub-cluster suppresses TGF-β-induced fibroblast activation in vitro [50]. miR-125b-5p negatively regulates p53 protein expression in human neuroblastoma cells and lung fibroblasts [51]. Overexpression of miR-181b-5p inhibits apoptosis and promotes fibroblast proliferation by regulating the MEK/ERK/p21 signaling pathway [52]. But increasing the expression of miR-376b-3p, on the contrary, inhibits the proliferation of fibroblasts causing their premature aging [53]. miR-1202 promotes fibrosis by inducing TGF-β1 [54].

33) In future studies, they authors should consider quantifying cell viability using other methods (trypan blue/ fix,stain nuclei and count on cytation5) as well. Assays such as MTT and CCK8 measure the activity of certain enzymes in the cells which we assume to correlate to cell number – but some treatments themselves can induce the activity of these enzymes, therefore, additional validation could strengthen their data.

During slide preparation for the comet assay, cells were counted after exposure to chrysotile asbestos using trypan blue. Analysis was performed on 100 cells per slide. A decrease in viable cells after asbestos exposure was observed by trypan blue counting.

44) Keeping with the viability assays, it would be good to look at additional longer time points as well – you might see a more profound effect on viability at later times.

In this study, asbestos exposure was modelled to determine acute effects, with a maximum exposure time of 48 hours. In addition, further research is being carried out to assess the chronic effects of chrysotile dust on the human body. To facilitate this, blood samples were taken from workers in an asbestos mining company with a working experience of between 5 and 20 years have been collected. A scientific article will be written based on the results of the research.

Line 60-61 uses a double negative, which makes it hard to understand.

We have remedied this deficiency.

Due to its structure, asbestos has high fire resistance, electrical and thermal insulation, this material has been widely used in various industrial spheres. Currently, there are few industries that do not use some of asbestos's valuable properties. However, by 2019, asbestos will have been banned in 66 countries and regions around the world [8]. World asbestos production in 2020 was more than 1.2 million tonnes. The main producing countries are Russia, Kazakhstan, China and Brazil [9]. In terms of chrysotile asbestos deposits, Kazakhstan ranks fourth in the world. The only producer of chrysotile fibre in Kazakhstan is Kostanay Minerals JSC. During its 55 years of operation, the company has produced more than 17 million tonnes of chrysotile, which is exported to 20 countries [10].

Reviewer 2 Report

Comments and Suggestions for Authors

The article attempts to caracterise the underlying mechanisms of asbestos induced lung diseases - a theme particularly interesting as asbestos exposure (mostly historical for developed countries) is still a real risk factor across some age groups and considering the unfavorable prognostic of pleural mesothelioma. The article may be of interest to: oncologists, respiratory physicians, public health specialists and particularly fundamental researchers.  

Some suggestions on improving the article: 

- perhaps the title should be modified - ‘...normal human lung fibroblasts’ is potentially misleading 

as MRC5 is a fetal line; the HLF line is closer to ‘...normal human lung fibroblasts’ 

same observation for line 104 

Introduction – good general presentation but potentially not adequate for a fundamental research audience 

Some sentences need rephrasing 

‘There are three forms of exposure to asbestos. Direct exposure to asbestos dust from 

mining operations. Indirect impacts are generally observed…’ the second phrase lacks a verb 

line 376 ‘it is believed that due to the low content of iron impurities, chrysotile fibers do not cause oxidative cellular stress.’ - is debatable – perhaps best removed; iron mobilization is correlated with DNA strand breaks to some extent see Hardy, Aust, 1995 

lines 384-390 Since chrysotile asbestos is the safest form of asbestos fiber, this material is freely 

available and widely used as a building material. It can be purchased not only on an in- 

dustrial scale, but also in mass markets. And its low cost makes chrysotile a very popular 

material in the construction market in countries where there is no ban on the use of asbes- 

tos. 

The results of our research showed that, contrary to the belief that chrysotile asbestos 

is safe for human health, this type of asbestos fiber can cause negative cellular effects. 

Various mechanisms of cellular response to asbestos are known. 

Partially true but better rephrased – the general perception is that chrysotile is dangerous (less than amphiboles) – would be interesting to mention the shorter biological halftime in lung tissues as this could complement your concentration/effect data – in terms of bioaccumulation phenomena 

same for 483/485 

‘While chrysotile asbestos is generally considered safer than other forms of asbestos and is believed to lack carcinogenic properties, this topic remains a subject of debate within the scientific community’  

debatable and not a conclusion of the experimental data presented 

line 461 

… screening of the population for the development of these diseases is necessary – is debatable – better remove; screening for lung cancer is particularly difficult as no clear biological marker is currently accepted  

Discussion
Existing results should be extensively commented and interpreted

Conclusions 

Should be reformulated to focus on actual research findings – dose effect, potential mechanisms 

Typing errors 

line 345 has-miR-1228 

table 1 – please correct spelling Crocodilite (multiple) – also see line 411 

remove Cyrillic 'I' used as ‘and’

Comments on the Quality of English Language

Some rephrasing might be necessary

Author Response

Dear Reviewer,

Thank you for reviewing our manuscript and providing feedback on our work. We hope you find the revised version to be satisfactory and thank you again for your time and attention. We have taken your comments into consideration and made the necessary improvements to enhance the clarity and precision of our text. We would like to express our gratitude for your valuable input and highlight that we have made every effort to maintain the original meaning of the text.

- perhaps the title should be modified - ‘...normal human lung fibroblasts’ is potentially misleading as MRC5 is a fetal line; the HLF line is closer to ‘...normal human lung fibroblasts’ same observation for line 104.

We completely agree with your comment. That's why they completely removed any mention about «normal human lung fibroblasts’». We have therefore changed the title of our article «Molecular and cellular mechanism of action of chrysotile asbestos in MRC5 cell line»

We've made an adjustment to line 104

MRC5 cells were cultured in Dulbecco’s Modified Eagle Medium (DMEM) (Capricorn Scientific GmbH, Germany) with high glucose content, L-glutamine, 100-unit penicil-lin-streptomycin (Capricorn Scientific GmbH, Germany) and 10% Fetal Bovine Serum (FBS) Collected in South America (Capricorn Scientific GmbH, Germany) in a humidified incubator at 37°C with 5% CO2.

Introduction – good general presentation but potentially not adequate for a fundamental research audience

Some sentences need rephrasing 

‘There are three forms of exposure to asbestos. Direct exposure to asbestos dust from

mining operations. Indirect impacts are generally observed…’ the second phrase lacks a verb

We've made the necessary adjustments to the text.

Exposure to asbestos occurs in three ways. Firstly, asbestos dust from mining operations provides direct exposure. Second, workers in the construction and automotive industries are typically exposed indirectly. Thirdly, there is exposure to the environment: asbestos fibres can contaminate the environment, for example in building materials, and ultimately expose the general public.

Due to its structure, asbestos has high fire resistance, electrical and thermal insulation, this material has been widely used in various industrial spheres. Currently, there are few industries that do not use some of asbestos's valuable properties. However, by 2019, asbestos will have been banned in 66 countries and regions around the world

Fibers less than 3 μm in diameter (chrysotile 0.026 μm) have the highest probability of entering the alveolar space and being cleared by surfactant proteins or macrophages [11]. Asbestos fibers that are too long penetrate into the cells, where they interfere with phagocytosis and trigger the primary production of reactive oxygen species (ROS). ROS production is primarily mediated by the reduction of Fe3+ iron ions on the surface of asbestos fibers [12]. Active cellular inflammation, in which macrophages initiate a cascade of biochemical reactions, results in secondary ROS production. a cascade of biochemical reactions, culminating in inflammation. (Figure 1) Gualtieri A outlines the diverse cellular impacts of asbestos exposure (including chrysotile), which include the formation of reactive oxygen species (ROS), the release of growth factors (TGF-β), the activation of p53, the activation of the Nalp3 inflammasome, the release of cytokines (TNF-α), the activation of transcription factors (AP-1, NF-κB) and the production of receptor tyrosine kinases (RTK) [13]. One consequence of oxidative stress is mtDNA release [14], which initiates a series of inflammatory responses leading to the development of specific diseases such as lung cancer [15].

Line 376 ‘it is believed that due to the low content of iron impurities, chrysotile fibers do not cause oxidative cellular stress.’ - is debatable – perhaps best removed; iron mobilization is correlated with DNA strand breaks to some extent see Hardy, Aust, 1995 

We've made the necessary adjustments to the text.

Different forms of asbestos have varying degrees of toxicity to the body: crocidolite is 500 times more toxic than chrysotile. Chrysotile asbestos, due to its structure, is considered the safest form of asbestos fiber. Currently, 95% of asbestos used and produced is chrysotile [26].

Lines 384-390 Since chrysotile asbestos is the safest form of asbestos fiber, this material is freely 

available and widely used as a building material. It can be purchased not only on an industrial scale, but also in mass markets. And its low cost makes chrysotile a very popular material in the construction market in countries where there is no ban on the use of asbestos. 

The results of our research showed that, contrary to the belief that chrysotile asbestos is safe for human health, this type of asbestos fiber can cause negative cellular effects. Various mechanisms of cellular response to asbestos are known. Partially true but better rephrased – the general perception is that chrysotile is dangerous (less than amphiboles) – would be interesting to mention the shorter biological halftime in lung tissues as this could complement your concentration/effect data – in terms of bioaccumulation phenomena 

We've made the necessary adjustments to the text.

Chrysotile asbestos is readily available and widely used in the construction industry because it is considered the least harmful form of asbestos fiber. It is available not only on an industrial scale, but also in mass markets. In countries where asbestos is not banned, chrysotile is a preferred material due to its affordability.

Research suggests that this type of asbestos fiber can have harmful effects on cellular functions, contrary to the belief that chrysotile asbestos is safe for human health. Several mechanisms have been identified for the cellular response to asbestos. The infiltration of asbestos fibers into cells leads to increased levels of reactive oxygen species (ROS). This causes oxidative stress, which in turn damages cellular components and DNA. As a result, a large number of signaling systems may be activated, leading to cell death or fibrosis and, subsequently, to carcinogenesis [28].

Same for 483/485

‘While chrysotile asbestos is generally considered safer than other forms of asbestos and is believed to lack carcinogenic properties, this topic remains a subject of debate within the scientific community’ debatable and not a conclusion of the experimental data presented

We've made the necessary adjustments to the text.

Chrysotile asbestos is generally considered to be a less dangerous form of asbestos and is not considered to be carcinogenic.

Line 461 

… screening of the population for the development of these diseases is necessary – is debatable – better remove; screening for lung cancer is particularly difficult as no clear biological marker is currently accepted  

We've made the necessary adjustments to the text.

Screening the population for lung cancer is particularly difficult because there is currently no clear biological marker. As a result, the diagnosis of lung cancer is often only at a late stage, when the disease is already advanced.  One promising marker for liquid biopsy is microRNAs.

Discussion
Existing results should be extensively commented and interpreted

We've made the necessary adjustments to the text.

Conclusions 

Should be reformulated to focus on actual research findings – dose effect, potential mechanisms 

We have completely rewritten the conclusion, hopefully we have conveyed the meaning and idea of the article as accurately as possible.

Typing errors

line 345 has-miR-1228 

We've made the necessary adjustments to the text.

We did not find any significant differences in the expression of hsa-miR-1228 after treating cells with different doses of chrysotile asbestos at specified time intervals (Figure 7).

table 1 – please correct spelling Crocodilite (multiple) – also see line 411 

Crocidolite induced an increase in ROS within 30 minutes of cell treatment and a further increase in ROS within 12 hours [28].

remove Cyrillic 'I' used as ‘and’

We've made the necessary adjustments to the table 1.

Reviewer 3 Report

Comments and Suggestions for Authors

The current article presents the effects of chrysotile asbestos on the MRC5 cell line. The authors emphasize the adverse effects of chrysotile asbestos using different method measurements of ROS, Comet Assay,  MTT assay, and qPCR. The study showed that Chrysotile leads to free radicals generation and increased oxidative stress in cells, mediates a high level of extracellular mitochondrial DNA, and thus exercises its genotoxic and cytostatic effects in the used diploid cell line.

The submitted manuscript is well-written, and the presented results are supported and proven by multiple analyses. The discussion is comprehensive, and the conclusion fits perfectly with the main scientific presentation in the article.

Author Response

The current article presents the effects of chrysotile asbestos on the MRC5 cell line. The authors emphasize the adverse effects of chrysotile asbestos using different method measurements of ROS, Comet Assay,  MTT assay, and qPCR. The study showed that Chrysotile leads to free radicals generation and increased oxidative stress in cells, mediates a high level of extracellular mitochondrial DNA, and thus exercises its genotoxic and cytostatic effects in the used diploid cell line.

The submitted manuscript is well-written, and the presented results are supported and proven by multiple analyses. The discussion is comprehensive, and the conclusion fits perfectly with the main scientific presentation in the article.

We thanks the Reviewer for the positive comments
